# Self-supervised Hierarchical Visual Reasoning with World Model

**Yuanfei Xu** [1]   **Lin Liu** [1]   **Wengang Zhou** [1 2]   **Mingxiao Feng** [2]   **Houqiang Li** [1 2]

## Abstract

3D open-world environments with adversarial opponents remain a core challenge for reinforcement learning due to their vast state spaces. Effective reasoning representations are essential in such settings. While existing self-supervised visual foresight reasoning approaches often suffer from multi-step error accumulation, many recent studies resort to injecting domain-specific knowledge for more stable guidance. Our key insight is that the photorealistic fidelity of visual reasoning representations is secondary; what truly matters is providing informative, task-relevant signals. To this end, we propose ResDreamer, a hierarchical world model in which each higher-level layer is trained to reconstruct the residuals of the layer below. This design enables progressive abstraction of increasingly sophisticated world dynamics and fosters the emergence of richer latent representations. Drawing inspiration from the "Bitter Lesson", ResDreamer trains its reasoning representations in a purely self-supervised manner. The higher-level residual representations are used to modulate lower-level predictions, allowing the world model to scale effectively with only linearly increasing cross-layer communication costs. Experiments show that ResDreamer achieves state-of-the-art sample efficiency and parameter efficiency. This scalable hierarchical visual foresight reasoning architecture paves the way for more capable online RL agents in open-ended, dynamic environments. The code is accessible at https://github.com/XuYuanFei01/ResDreamer.

## 1. Introduction

In interaction or combat scenarios, task objectives are relevant to dynamic elements or proactive agents. This introduces a highly dynamic and uncertain environment evolution. The vast state space of a 3D open-ended environment exacerbates this challenge. The agent must construct an internal world representation based on partial information and make decisions accordingly.

World models have significantly advanced the frontiers of reinforcement learning (RL) (Schrittwieser et al., 2020; Robine et al., 2023; Zhang et al., 2023; Alonso et al., 2024). DreamerV3 (Hafner et al., 2025) achieves strong generalization across over 150 diverse tasks using a single set of hyperparameters. Robotic World Model (Li et al., 2025a) enables stable prediction of 100+ proprioceptive observation steps. BOOM (Zhan et al., 2025) integrates world models with sampling-based online planning, attaining state-of-the-art (SOTA) sample efficiency on high-dimensional locomotion tasks. However, existing model-based RL (MBRL) methods primarily rely on latent-space look-ahead search or model predictive control. Pixel-level reconstruction is rarely exploited as a reasoning representation, largely because it provides no additional information beyond latent representations and suffers from compounding error accumulation.

Reasoning representations are critical for long horizon tasks. Large Language Models (LLMs) can provide advanced and interpretable reasoning representations, such as decomposed sub-task prompts and policy-as-code formulations. Recent LLM-powered embodied agents tailored to the Minecraft environment include JARVIS-1 (Wang et al., 2023b), MC-Planner (Wang et al., 2023c), and RL-GPT (Liu et al., 2024). However, despite recent efforts to mitigate the latency of embodied Chain-of-Thought (CoT) reasoning (Zheng et al., 2025; Duan et al., 2025), language-grounded CoT remains poorly suited to high-frequency, dynamic environments due to its inherent inference overhead and lower temporal resolution.

Embodied visual reasoning representations—such as affordance maps (Chu et al., 2025), visual grounding (Li et al., 2024b; Chen et al., 2025b), and goal images (Zhao et al., 2025)—have promised superior task performance. However, they typically rely on domain-specific priors and environment-anchored annotations. In contrast, large-scale

[1]Department of Electronic Engineering and Information Science, University of Science and Technology of China, Hefei, Anhui, China [2]Institute of Artificial Intelligence, Hefei Comprehensive National Science Center, Hefei, Anhui, China. Correspondence to: Mingxiao Feng <fengmx@iai.ustc.edu.cn>, Houqiang Li <lihq@ustc.edu.cn>.

*Proceedings of the 43rd International Conference on Machine Learning*, Seoul, South Korea. PMLR 306, 2026. Copyright 2026 by the author(s).

domain-general video generation models trained in an unsupervised manner, such as Genie (Bruce et al., 2024), WAN (Wan et al., 2025) and Cosmos (NVIDIA et al., 2025), generally comprise over 10B parameters. Their inference latency and deployment costs remain high for embodied online visual reasoning.

Neuroscience evidence suggests that the biological neural signals encode prediction error rather than raw sensory input (Rao & Ballard, 1999; Hosoya et al., 2005). Visual neurons employ a dynamic predictive coding strategy to filter out predictable components from the visual stream, transmitting only unexpected surprise or "report valuable" stimuli (Kok & de Lange, 2015).

Building on these insights, we introduce ResDreamer, a hierarchical world model that employs residually connected visual planning representations. By modeling visual reconstruction residuals, each higher-level layer not only builds a richer, more comprehensive internal world representation but also refines the foresight of lower layers through residual reasoning, thereby providing more informative reasoning representation.

We adopt a highly lightweight parameter regime: ResDreamer operates with only 50–200 million parameters yet successfully solves long-horizon combat tasks in Minecraft open-world environments that feature complex battle mechanics. Our investigation centers on the novel residually connected visual reasoning representation we propose; accordingly, we incorporate no language-conditioned modules. We assess the method's effectiveness via online RL success rates on these combat tasks.

In summary, the major contributions of this work are:

- We introduce a hierarchical world model architecture for representation learning, in which upper layers learn progressively more advanced world dynamics from the residuals of lower layers. This design paves the way for the "ResNet era" of world models where residual connections enable deeper hierarchies.

- We propose a residual-enhanced visual reasoning representation. By modulating visual foresight with normalized upper-layer residual predictions, we deliberately forgo photorealistic fidelity and instead emphasize unexpected visual stimuli, thereby delivering highly informative reasoning signals

- ResDreamer achieves state-of-the-art sample efficiency and parameter efficiency in online RL settings, and it is the only method to exhibit non-near-zero success rates on high-difficulty tasks such as hunting Shulkers. Ablation studies further isolate the contributions of foresight rollouts and residual modulation.

## 2. Related Work

**Imagination-driven MBRL**. Recurrent world dynamic models facilitate representation, simulation and policy improvement in MBRL (Ha & Schmidhuber, 2018). MuZero (Schrittwieser et al., 2020) conducts Monte Carlo tree search in the latent space by the learned state space model. DreamerV3 (Hafner et al., 2025) outperformed expert models tuned for specific domains and, for the first time, successfully collected diamonds from scratch in Minecraft. LS-Imagine (Li et al., 2024a) breaks the limitations of single-step reasoning and uses the affordance map to trigger the cross-step jump prediction. It simulates jumping to the vicinity of high return targets in the future by magnifying specific areas in the observed image. In visual MBRL, transformer-based architectures (Micheli et al., 2022; Robine et al., 2023; Zhang et al., 2023) and diffusion models (Alonso et al., 2024) have emerged as particularly effective paradigms for world modeling, offering enhanced expressivity and sample efficiency. Although these methods support look-ahead search via Monte Carlo Tree Search (MCTS), none of them are designed to provide reasoning representations that supply additional visual guidance. ResDreamer follows the imagination-driven MBRL paradigm, in which the actor-critic is trained purely on imagined trajectories. This design completely avoids policy distributional divergence. Consequently, we can leverage a massive replay buffer as a rich self-supervised signal for world model representation learning, while all policy and value updates remain strictly online. What sets our approach apart is that it naturally constructs and utilizes a hierarchical reasoning representation built directly upon the residuals of sensory signals.

**Embodied Reasoning Representation**. Embodied reasoning is transitioning from high-latency, high-performance paradigms toward compact and real-time approaches. For instance, Fast-ThinkAct (Huang et al., 2026) introduces verbalizable latent planning, where a compact latent CoT is distilled from a teacher model via teacher-student distillation, achieving up to 89.3% reduction in inference latency. In highly compositional dynamic scenarios, object-centric reasoning representations have proven effective in improving prediction accuracy for dynamic entities and enhancing agents' ability to interact intelligently with them. Meta AI's Vision-Language World Model (VLWM) (Chen et al., 2025a) exemplifies an alternative reasoning pathway that avoids pixel-level prediction altogether: it uses natural language as an abstract representation of world evolution, achieving extreme semantic compression through structures such as the Tree of Captions. Beyond latent-space reasoning, leveraging domain knowledge to infer goal states can also be beneficial. Puppeteer (Hansen et al., 2025a) directly employs expert trajectories from human motion capture data to train high-level goal synthesis for low-level whole-body humanoid controllers. COVR (Xia et al., 2026) proposes a

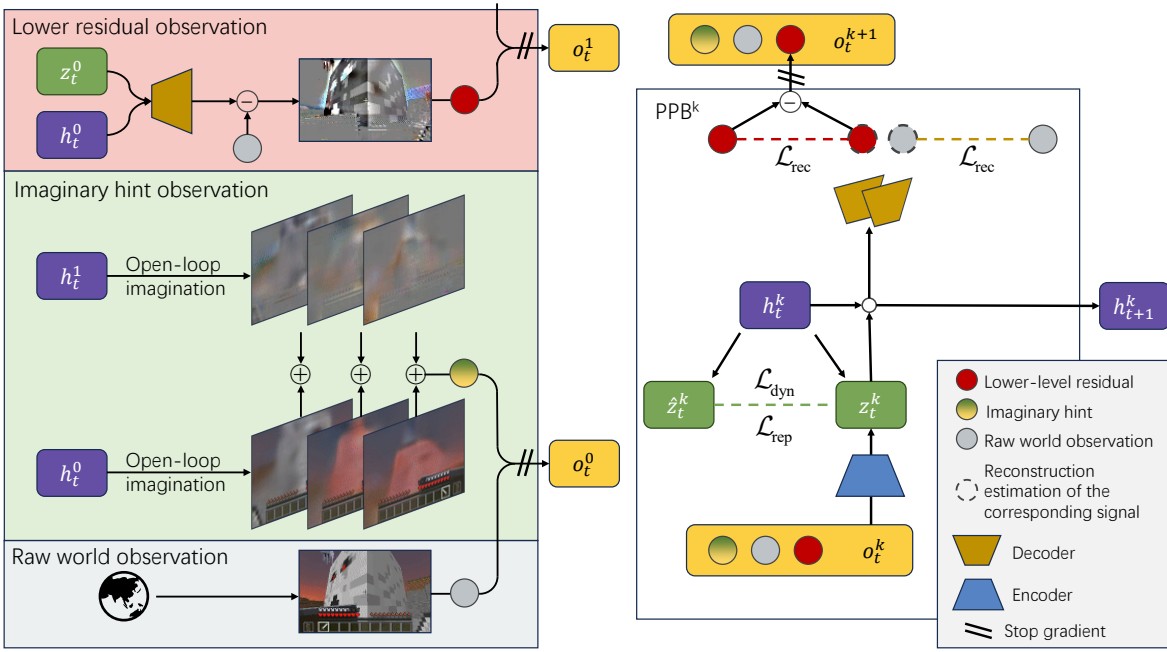

*Figure 1.* Overview of ResDreamer, a model-based RL algorithm based on hierarchical world model. The left side shows the structure of enhanced visual observations. Adjacent world model layers communicate by residual and predictive signal within the enhanced observation. The right side shows the modules and training process of the k-th layer world model. The Encoder reads enhanced visual observations and gives the posterior $z_t^k$. The dynamic predictor learns to estimate $z_t^k$ with $\hat{z}_t^k$ without accessing the observation. The sequence model updates internal state $h_t^k$ by $z_t^k$. The Decoder reconstructs the observation signal which generates reconstruction loss and residual visual signal for upper layer.

bidirectional learning framework that accelerates RL training by using prior guidance from vision-language models (VLMs), while high-quality trajectories collected by the RL agent during environment interaction are used to fine-tune the VLM. In contrast, our proposed ResDreamer adopts a reasoning representation that is both highly efficient for real-time inference. Its latent rollout and pixel-space foresight facilitates efficient reasoning and feature extraction. Moreover, pixel-space signals offer interpretability and the additive property of residuals. Within a lightweight 50-200M parameter scale, we demonstrate that even blurry yet task-relevant visual foresight provides substantial benefits.

**Compared to Hierarchical RL**. Hierarchical reinforcement learning (HRL) is widely regarded as a promising approach for mitigating exploration stagnation induced by sparse rewards in long-horizon tasks. A central focus in HRL is sub-goal discovery. Classic goal-conditioned policies obtain sub-goals either by extracting them from successful trajectories in the replay buffer (e.g., HER) or by treating sub-goals as high-level policies in a broader sense (e.g., HIRO). Another key direction involves learning world dynamics models at multiple temporal scales. For example, THICK (Gumbsch et al., 2024) adaptively discovers larger temporal abstractions by guiding lower-level world models to sparsely update their partial latent states. Recent

work has explored more efficient sub-goal discovery algorithms and representations (Hafner et al., 2022; Hamed et al., 2024; Hansen et al., 2025b). The motivation of ResDreamer differs fundamentally from these prior HRL approaches. ResDreamer primarily focuses on layered hierarchical representation learning of world dynamics and leverages the residual-enhanced reasoning representations to boost performance of RL tasks.

## 3. Method

In this section, we present the details of ResDreamer. We introduce ResDreamer from the perspectives of representation learning and behavior learning. First, we describe the basic module of each layer in our Hierarchical Recurrent State-Space Model (HRSSM). Next, we present our primary innovation in representation learning architecture, namely the enhanced observation through residual connection. Finally, we formalize the loss functions and the overall training algorithm.

### 3.1. Hierarchical World Model

We implement the HRSSM based on Predictive Processing Blocks (PPBs). Predictive Processing or Predictive coding is a paradigm to explain the hierarchical reciprocal connection

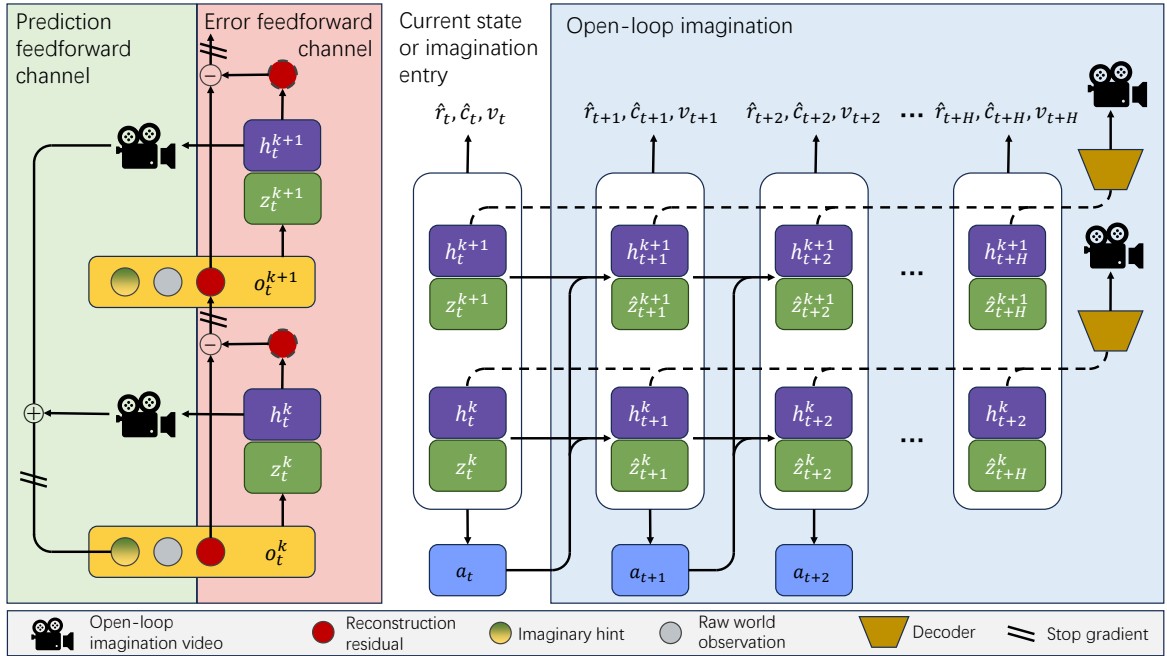

*Figure 2.* The information channel between world model layers is bidirectional. Only reconstruction error and modulated foresight images are transmitted between layers, with no gradients being passed. On one hand, each layer of the PPB generates predictions about the external world and transmits visual planning representations to lower layers. On the other hand, the PPB treats low-level residuals as self-supervised learning signals to obtain a more complete inner representation.

of the cortex (Huang & Rao, 2011).

In the k-th layer block $PPB^k$, recurrent state contains the deterministic state $h_t^k$ and the stochastic state $z_t^k$. The sequence model is used to represent the state transitions conditioned by action taken. The Encoder extracts useful information from the new input observations to guide the recurrent state update, while the Predictor attempts to predict the stochastic state without accessing the observations.

$$PPB^k \begin{cases} \text{Sequence model:} & h_t^k = S_\phi \left( z_{t-1}^k, h_{t-1}^k, a_{t-1} \right) \\ \text{Encoder:} & z_t^k \sim q_\phi \left( z_t^k \mid h_t^k, o_t^k \right) \\ \text{Predictor:} & \hat{z}_t^k \sim p_\phi \left( \hat{z}_t^k \mid h_t^k \right) \\ \text{Decoder:} & \hat{o}_t^k \sim D_\phi \left( \hat{o}_t^k \mid h_t^k, z_t^k \right). \end{cases}$$
(1)

where $\hat{z}_t^k$ is the predicted stochastic state, $o_t^k$ and $\hat{o}_t^k$ are true and reconstructed observations. Layer index $k = 0, 1, \cdots L - 1$ and $L$ is the number of HRSSM layers.

### 3.2. Visual Hint Structure and Residual Modeling

Figure 1 gives an overview of ResDreamer, a hierarchical world model in which layers communicate through error feedback and predictive visual hints. This section elaborates on the forms of information exchange between the world model layers. ResDreamer is characterized by progressive residual learning of sensory signals and image foresight corrected by residual prediction.

Sensory reconstruction error is fed into the higher-level world model for residual learning. The **lower residual observation** is given by

$$o_{\text{res}}^1 = \text{Norm}^1 \left( o_{\text{raw}} - \hat{o}_{\text{raw}} \right),$$
$$o_{\text{res}}^k = \text{Norm}^k \left( o_{\text{res}}^{k-1} - \hat{o}_{\text{res}}^{k-1} \right),$$
(2)

where $k = 2, 3, \cdots, L - 1$, the omitted time indices are all $t$, and the same applies hereafter. The bottom layer only has environmental observations $o_{\text{raw}}$ and has no residual observations. $\text{Norm}^k(\cdot)$ computes the mean and variance across the pixel dimension and updates them with exponential moving average.

It is worth noting that any layer of the well-trained PPB can rollout latent trajectories by replacing the posterior with the prior. This means that as long as PPB is trained to model the lower residual, it can perform open-loop reasoning and correct the visual reasoning representations at the lower level. The **imaginary hint observation** is given by

$$o_{\text{imag}}^0 = \left\{ \hat{o}_{\text{raw}} + \hat{o}_{\text{res}}^1 \right\}_{t:t+H},$$
$$o_{\text{imag}}^k = \left\{ \hat{o}_{\text{res}}^k + \hat{o}_{\text{res}}^{k+1} \right\}_{t:t+H},$$
$$o_{\text{imag}}^{L-1} = \left\{ \hat{o}_{\text{res}}^{L-1} \right\}_{t:t+H}.$$
(3)

where $k = 1, 2, \cdots, L-2$, the subscript $\{\cdot\}_{t:t+H}$ stands for concatenation of reconstructed open-loop imagination for

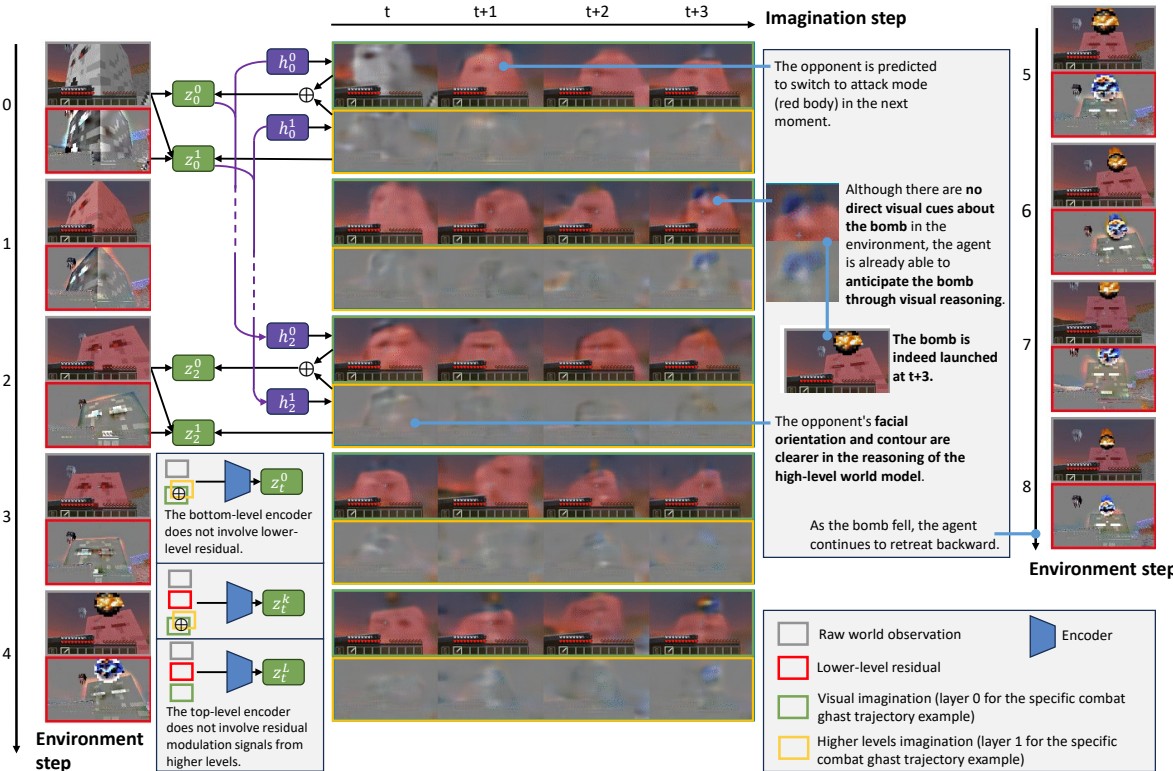

*Figure 3.* Visualization of residual-modulation mechanism of a two-layer world model example. The trajectory clip shows a ghost switching to attack mode (red body) and launching a bomb. The agent is able to anticipate the bomb before it appears through visual reasoning. As the bomb fell, the agent continues to retreat.

future time $t, t+1, \cdots, t+H-1$. Specifically, if the raw image shape is $(h, w, 3)$, then $\{\hat{o}^k_{\text{res}}\}_{t:t+H}$ and $\{\hat{o}^k_{\text{imag}}\}_{t:t+H}$ decoded from a $H$ steps latent trajectory both have the shape of $(H, h, w, 3)$. They are added by residual connections and concatenated along the channel axis into shape $(h, w, 3H)$.

The **imaginary hint observation** utilizes the hierarchical structure of the world model to build visual reasoning representation. The foresight is blurry but informative and expressive. See Figure 3 which shows the raw observation and imaginary hint observation on world model bottom layer while the agent is combating a ghost.

The observation expanded by lower-level residuals and upper-level predictions is the key to our hierarchical world model:

$$o^k_t = \text{sg}\left(\left\{o^k_{\text{imag}}, o_{\text{raw}}, o^k_{\text{res}}\right\}_t\right). \qquad (4)$$

where $\text{sg}(\cdot)$ is stop gradient operation. In our implementation, all the observations are concatenated along the channel axis. Eventually, a complete enhanced observation tensor is formed with the shape of $(h, w, 3H + 6)$.

The **imaginary hint observation** is merely additional guidance or enhancement for the encoder. Only the **lower residual observation** and **environmental input** are recon-

structed by the decoder and provide reconstruction loss during the training $\hat{o}^k_t = \text{sg}\left(\{\hat{o}_{\text{raw}}, \hat{o}^k_{\text{res}}\}_t\right)$. The complete process of constructing enhanced observations from the bottom layer to the top layer and updating the recursive state in sequence is shown in Algorithm 1.

At this point, we have established the feedforward and feedback information channels of the hierarchical world model based on the enhanced observation (see Figure 2). This architecture combines the bandwidth advantage of inter-layer communication and the computational efficiency advantage within layers.

The visual hint does incur necessary computational cost, but from the perspective of parameter scale, the above architecture introduces almost no overhead. Within each layer, although the number of image channels has significantly increased due to the addition of video hint, the distribution of the visual hint is highly matched with the original image distribution benefits from the residual modeling, thus allowing for the sharing of major convolutional features. Therefore, in practice, we have not expanded the depth of the encoder and dimensions of stochastic state compared to DreamerV3 (Hafner et al., 2025).

## 3.3. World Model and Behavior Learning

In the context of RL, the final goal is to improve the policy. The actor-critic method is employed for policy optimization and state value learning.

$$
\begin{aligned}
\text{Actor:} &\quad a_t \sim \pi_\theta \left( a_t \mid s_t \right) \\
\text{Critic:} &\quad v_t \sim v_\psi \left( v_t \mid s_t \right)
\end{aligned}
\tag{5}
$$

The world environment generously provides continuous stream of sensory signals. Reconstructing sensory inputs serves as a critical training signal for world models. This drives the model to encode as much environmental information as possible in deterministic state.

$$
\mathcal{L}_{\text{rec}}^k (\phi) = - \ln p_\phi \left( o_{\text{raw}} \mid s_t^k \right) - \ln p_\phi \left( o_{\text{res}}^k \mid s_t^k \right).
\tag{6}
$$

The stochastic state serves as the information channel between observation and latent state representation. The enforced sparsity makes the stochastic state more predictable, while the representation loss ensures that the posterior distribution converges to a more predictable distribution.

$$
\begin{aligned}
\mathcal{L}_{\text{dyn}}^k(\phi) &= \text{KL} \left[ \text{sg} \left( q_\phi \left( z_t^k \mid h_t^k, o_t^k \right) \right) \,||\, \quad p_\phi \left( z_t^k \mid h_t^k, \right) \right] \\
\mathcal{L}_{\text{rep}}^k(\phi) &= \text{KL} \left[ \quad q_\phi \left( z_t^k \mid h_t^k, o_t^k \right) \,||\, \text{sg} \left( p_\phi \left( z_t^k \mid h_t^k \right) \right) \right]
\end{aligned}
\tag{7}
$$

Additional prediction heads perform reward modeling $\hat{r}_t \sim p_\phi \left( \hat{r}_t \mid s_t \right)$ and episode-continuation flag $\hat{c}_t \sim p_\phi \left( \hat{c}_t^k \mid s_t \right)$ are also trained in a self-supervised manner:

$$
\mathcal{L}_{\text{heads}}(\phi) = - \ln p_\phi \left( r_t \mid s_t \right) - \ln p_\phi \left( c_t \mid s_t \right)
\tag{8}
$$

The actor-critic is trained purely on imagined trajectories. We compute the bootstrapped $\lambda$-return $R_t^\lambda$ to train the critic. $R_t^\lambda$ accounts for $r_t$ within the trajectory horizon $T$ and incorporates the critic's expected value for returns beyond the horizon.

$$
\begin{aligned}
\mathcal{L}(\psi) &= - \sum_{t=1}^{T} \ln p_\psi \left( R_t^\lambda \mid s_t \right) \\
R_t^\lambda &= \begin{cases} r_t + \gamma c_t \left( (1 - \lambda) v_t + \lambda R_{t+1}^\lambda \right), & t < T \\ \mathbb{E} \left[ v_\psi \left( \cdot \mid s_t \right) \right], & t = T \end{cases}
\end{aligned}
\tag{9}
$$

The actor learns to maximize returns with entropy regularizer:

$$
\begin{aligned}
\mathcal{L}(\theta) = &- \sum_{t=1}^{T} \frac{R_t^\lambda - \text{sg} \left( v_\psi \left( s_t \right) \right)}{\max(1, S)} \log \pi_\theta \left( a_t \mid s_t \right) \\
&+ \eta \text{H} \left[ \pi_\theta \left( a_t \mid s_t \right) \right]
\end{aligned}
\tag{10}
$$

where $S = \text{EMA} \left( \text{Per} \left( R_t^\lambda, 95 \right) - \text{Per} \left( R_t^\lambda, 5 \right) \right)$ is the scale from the 5% quantile to the 95% quantile maintained by the exponential moving average. The scale factor $S$ is used to normalize the return and is clipped to be no less than 1 to prevent noise amplification. Overall, we follow the official DreamerV3 (Hafner et al., 2025) hyperparameters and implementation details. Benefiting from DreamerV3's robust generalization capabilities, ResDreamer can be trained across diverse tasks without hyperparameter tuning.

Residual Enhanced Visual Observation is the main innovation of this work. To visually demonstrate the structure of this visual foresight and the planning information it provides, we visualize the bottom layer observation in Figure 3.

## 4. Experiments

Engaging in combat within open-ended 3D worlds poses substantial challenges, requiring robust terrain understanding, effective use of weapons and defensive tools, and real-time anticipation of adversarial enemy movements and behaviors. We evaluate ResDreamer on five combat tasks from the MineDojo benchmark (Fan et al., 2022) detailed in Table 3. These tasks span a range of difficulty levels and involve fighting various hostile entities under diverse initial inventories and lighting setups.

The only non-trivial hyperparameter introduced by ResDreamer is the rollout horizon $H$ for image foresight predictions. To assess its impact, along with the complementary time stride $D$ (the interval between predicted frames), we conduct a sensitivity analysis on a suite of visual continuous control tasks from the DeepMind Control Suite (DMC) (Ortiz et al., 2024) with pixel observations.

### 4.1. Main Comparison

We evaluate all methods using the success rates across tasks. The training curves in Figure 4 and bar charts in Figure 8 demonstrate that ResDreamer achieves superior sample efficiency as a hierarchical model-based RL method. To enable a fair assessment of parameter efficiency, we test ResDreamer under two configurations:

- ResDreamer (100M×2): a two-layer hierarchical model with approximately 100 million parameters per layer (total 200M). This variant exhibits the strongest sample efficiency and fastest convergence across all evaluated tasks.

- ResDreamer (50M×2): a lighter two-layer model. It surpassed the average performance of DreamerV3 with only 84% of the parameter size.

The detailed parameter sizes, hyperparameters, and compute budgets are shown in Table 2. All the baselines use the

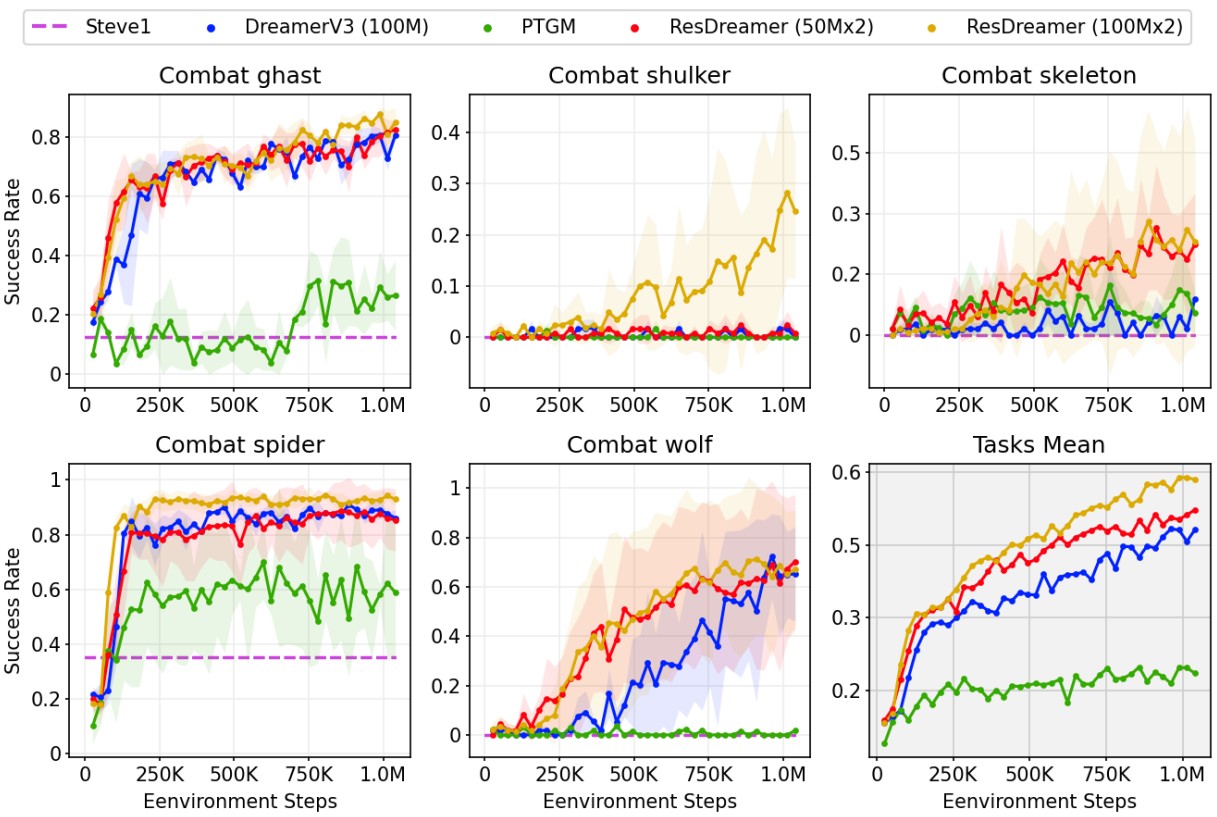

*Figure 4.* Comparison of ResDreamer against Steve-1 (Lifshitz et al., 2023), DreamerV3 (Hafner et al., 2025), PTGM (Yuan et al., 2024). We introduce the compared models in Appendix E. Results of transformer based MBRL method IRIS (Micheli et al., 2022) is presented in Appendix D.1

default configuration of the official implementation. Further details are provided in Appendix A.

ResDreamer (100M×2) is the only method among the evaluated baselines that successfully solves the high-difficulty Shulker combat task within $1 \times 10^6$ environment steps. The hostile mob Shulker launches a guided projectile that causes prolonged levitation and subsequent fall damage. This poses complex and challenging dynamic interaction mechanisms for the agent. Our analysis shows that the residual-modulated visual reasoning representation in Res-Dreamer provides critical assistance in mastering this interaction. The reasoning representation supplies modulated foresight images that enrich the observation stream with predictive signals about impending changes (e.g., incoming projectiles, see also Figure 3), making the input more informative for decision-making.

### 4.2. Model Analysis

The foresight rollout and residual connection mechanism are the key components of ResDreamer. We present ablation study to isolate the contribution from each design.

#### 4.2.1. ABLATION STUDY

The residual connections in ResDreamer enable the flow of both predictive signals and reconstruction errors across hierarchical layers, forming a key mechanism for enhanced visual reasoning. Figure 5 presents ablation results comparing the standard ResDreamer against several alternative configurations.

**ResDreamer 50M×3**. We extend the main two-layer model (50M×2) to three layers while maintaining a similar per-layer parameter budget. The three-layer variant achieves a higher mean task success rate than the two-layer version. This demonstrates that ResDreamer's residual hierarchy provides an effective and scalable pathway for world models.

**ResDreamer Only Residual**. In this setup, imaginary hint observations consist exclusively of upper-layer residual signals, excluding the current layer's own predictive reconstruction. Although the current layer's latent state already encompasses complete information from open-loop predictions, we find that blending predictive reconstruction with residual signals yields superior performance.

**ResDreamer W/O Residual**. We remove residual connec-

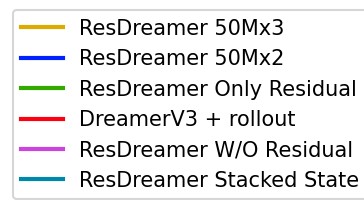

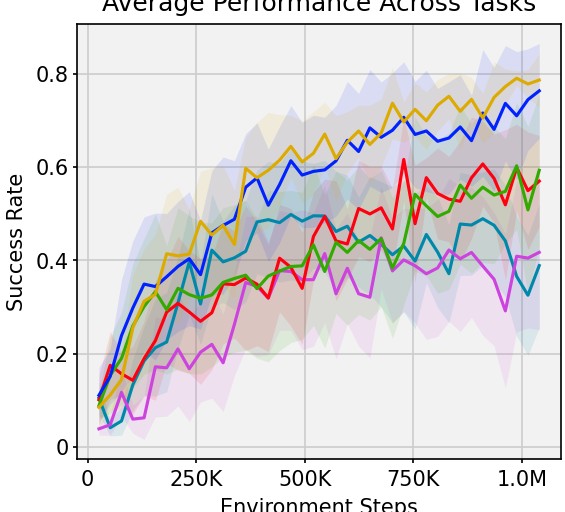

*Figure 5.* ResDreamer ablation study results.

tions entirely, so actor-critic and prediction heads only access upper layer latent state for additional information. Task performance drops markedly, underscoring that residual-modulated visual foresight is the core component that improves performance.

**ResDreamer Stacked State**. The actor, critic, and prediction heads in ResDreamer are conditioned on the stacked latent states of all layers. Despite the stacked states theoretically containing richer recursive information, performance declines under 1M environment steps. We assume that this degradation arises because the distribution of lower-layer residuals shifts during training, causing instability in upper-layer representations before full convergence. Future work could explore longer training regimes or adaptive normalization to mitigate this effect.

**DreamerV3 + rollout**. We augment a standard single-layer DreamerV3 baseline by adding online-computed visual foresight rollouts and reconstructions as additional grounded reasoning observations. Despite this enhancement, performance remains below ResDreamer, further validating the synergistic benefits of the hierarchical architecture combined with residual connections.

These ablations collectively isolate the contributions of residual modulation and hierarchical depth, confirming that both elements are essential for ResDreamer's superior per-

formance.

### 4.2.2. FORESIGHT HORIZON SENSITIVITY ANALYSIS

We evaluate ResDreamer using foresight horizons $H = 4, 8, 16$ and strides $D = 1, 2, 4$ respectively, keeping the total number of predicted frames fixed at 4. Experiments are conducted on visual continuous control suite DMC Vision.

As shown in Figure 6, longer horizon and larger stride eventually converge to higher average performance. However, excessively long horizons can slow early-stage convergence on some tasks due to initially noisy predictions.

ResDreamer consistently outperforms DreamerV3 under different horizon configurations. In DMC Hopper Hop, $H = 16$ achieves the best results, as 16 steps align closely with a full hopping cycle, enabling better anticipation of balance-critical dynamics.

Overall, longer and coarser foresight proves more informative for complex dynamics after sufficient training, though shorter horizons may accelerate early-stage learning.

## 5. Conclusion

In this paper, we present ResDreamer, a hierarchical world model that employs residually connected visual planning representations. By modeling reconstruction residuals, each layer passes only the novel, unexpected sensory signals upward, creating a bandwidth efficient information channel between layers. Residual rollouts from upper layers modulate the visual foresight at lower levels, enriching the predictive reasoning representation.

Extensive comparisons and ablation studies demonstrate that ResDreamer achieves superior sample efficiency and parameter efficiency in challenging online visual RL tasks. Critically, the synergy of hierarchical structure, residual modulation, and modulated image foresight proves substantially more effective than any subset of these components.

The primary limitation of the current approach is the fixed foresight horizon length. Longer static horizons incur higher computational cost, while overly short horizons may fail to capture sufficient long-range dynamics. Developing adaptive or learnable horizon mechanisms remains an important direction for future work.

Overall, ResDreamer's task-agnostic reasoning representation readily adapts to any visual RL scenario, establishing a powerful and broadly applicable baseline for imagination-driven MBRL across discrete and continuous action spaces. It defines the frontier of online RL agents in 3D open-ended dynamic interactive environments.

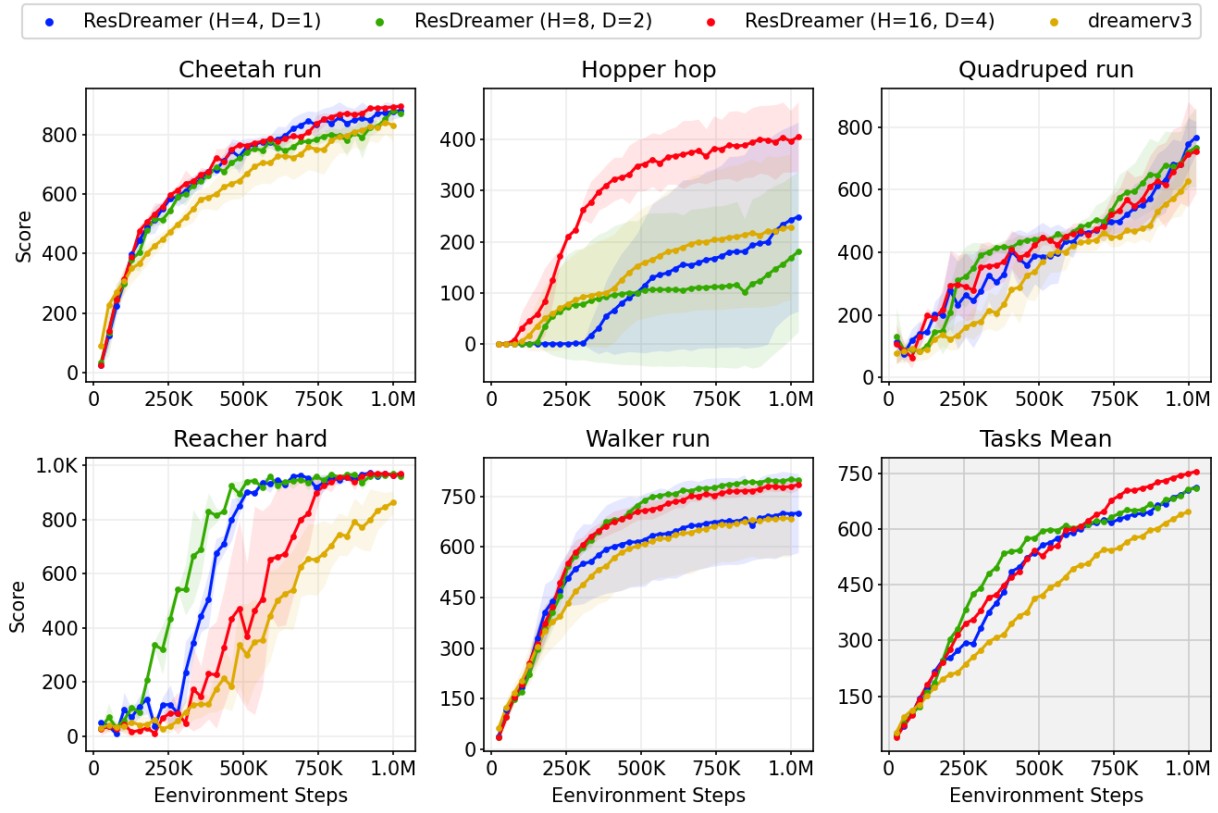

*Figure 6.* Comparison of ResDreamer with different foresight horizon on DMC Vision (Ortiz et al., 2024) continuous control suite.

## Acknowledgements

This work was supported by National Key RD Program of China under Contract 2022ZD0119802 and the Youth Innovation Promotion Association CAS. It was also supported by the GPU cluster built by MCC Lab of Information Science and Technology Institution and the Supercomputing Center of USTC.

## Impact Statement

This paper presents work whose goal is to advance the field of Machine Learning. There are many potential societal consequences of our work, none which we feel must be specifically highlighted here.

In this work we adhere to the code of ethics. This work does not involve human subjects, personal data, or sensitive information.

## Conflict of Interest

The authors declare that they have no known competing financial interests or personal relationships that could have appeared to influence the work reported in this paper.

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

## A. Reproducibility Statement & Detailed Hyperparameters

Our codebase is accessible at `https://github.com/XuYuanFei01/ResDreamer`. We base our implementation on the released official DreamerV3 codebase `https://github.com/danijar/dreamerv3`. The hyperparameter details are listed in Table 1.

*Table 1.* Hyperparameter settings.

| Hyperparameter | Value | Hyperparameter | Value |
|---|---|---|---|
| **Training** | | | |
| Learning rate | $4 \times 10^{-5}$ | Slow value network update rate | 0.02 |
| Environment samples | 4 | Value horizon | 333 |
| Replay batch size | 16 | Discount factor ($\gamma$) | 1-1/333 |
| Replay batch length | 64 | Optimizer | Adam |
| Buffer size | $1 \times 10^{6}$ | Train ratio | 48 |
| **World Model** | | | |
| Imagination length | 15 | Return normalization | Moving (5%, 95%) |
| Reward loss scale | 1 | Dynamics loss scale | 1 |
| Value loss scale | 1 | Representation loss scale | 0.1 |
| Policy loss scale | 1 | Replay value loss scale | 0.3 |
| Reconstruction loss scale | 1 | | |
| **Actor-Critic** | | | |
| bootstrap $\lambda$-return $\lambda$ factor | 0.95 | Slow value regularization scale | 1 |
| Entropy coefficient ($\alpha$) | $3 \times 10^{-4}$ | Value distribution | symexp twohot |

In this paper, the CPU used is an Intel Core i9-14900K, and the GPU used is an NVIDIA GeForce RTX 5090. GPU VRAM consumption for the largest 100Mx2 configuration is under 29GB.

*Table 2.* ResDreamer and DreamerV3 world model settings.

| Configurations | DreamerV3 | ResDreamer (50Mx2) | ResDreamer (100Mx2) |
|---|---|---|---|
| Foresight horizon | 0 | 4 | 4 |
| Recurrent $h_t$ size | 6144 | 4096 | 6144 |
| Recurrent $z_t$ size | $32 \times 48$ | $32 \times 32$ | $32 \times 48$ |
| Hidden size | 768 | 512 | 768 |
| Encoder CNN channels | 48 | 32 | 48 |
| Decoder CNN channels | 32 | 32 | 32 |
| hierarchies | 1 | 2 | 2 |
| Total parameters | 109.5M | 92.0M | 192.7M |
| Total training hours | 6.2 | 12.3 | 14.5 |

## B. Algorithm Details

Figure 2 illustrates the data flow diagram of open-loop imagination and how it constructs enhanced visual observations during training. Our hierarchical model extends the process of updating the internal recurrent state based on observations. See Algorithm 1 for details. The sequence of environmental interactions stored in the replay buffer is utilized only for training the representational learning of the world model, while policy improvement relies exclusively on imagined trajectories. Consequently, the training pipeline and the environment interaction are entirely asynchronous. For a detailed description of the training pipeline, refer to Algorithm 2.

---

**Algorithm 1** Update the recurrent state of ResDreamer upon observation

---

**Input:** recurrent state $s_t$, raw observation $o_{\text{raw}}$.
**Output:** recurrent state $s_{t+1}$, world model losses $\mathcal{L}_{\text{dyn}}(\phi), \mathcal{L}_{\text{rep}}(\phi), \mathcal{L}_{\text{rec}}(\phi)$.
 1: Open-loop rollout imaginary state-action trajectory $\left\{\hat{s}^{0:L-1}, a\right\}_{t+1:t+H}$
 2: initiate $o_{\text{res}}^k$ with empty set.
 3: **for** each $k = 0, 1, \cdots, L-1$ **do**
 4:      Compute $o_{\text{imag}}^k$ with Eq. 3.
 5:      Compute $o_t^k$ with Eq. 4.
 6:      $z_t^k \leftarrow \text{sample}\left[q_\phi\left(z_t^k \mid h_t^k, o_t^k\right)\right]$. {Encoder }
 7:      $\hat{z}_t^k \leftarrow \text{sample}\left[p_\phi\left(z_t^k \mid h_t^k\right)\right]$. {Predictor }
 8:      Compute prediction loss $\mathcal{L}_{\text{dyn}}^k(\phi)$ and representation loss $\mathcal{L}_{\text{rep}}^k(\phi)$ with Eq. 7.
 9:      $h_{t+1}^k \leftarrow S_\phi\left(z_t^k, h_t^k, a_t^k\right)$. {Sequence model }
10:      Compute sensory signal reconstruction $\hat{o}_t^k = \left\{\hat{o}_{\text{raw}}^k, \hat{o}_{\text{res}}^k\right\}_t$. {Decoder}
11:      Compute reconstruction loss $\mathcal{L}_{\text{rec}}^k(\phi)$ with Eq. 6.
12:      Compute $o_{\text{res}}^k$ with Eq. 2.
13: **end for**
     return $s_{t+1}, \mathcal{L}_{\text{dyn}}(\phi), \mathcal{L}_{\text{rep}}(\phi), \mathcal{L}_{\text{rec}}(\phi)$.

---

**Algorithm 2** The training pipeline of ResDreamer

---

 1: initiate parameters $\phi, \theta, \psi$.
 2: initiate carried state $s_{\text{carry}}$.
 3: **while** not converged **do**
 4:      {World model representation learning }
 5:      Sample a environmental interaction sequence $\{o_{\text{raw}}, a\}_{0:T-1}$ from replay buffer.
 6:      **for** each $t = 0, 1, \cdots, T-1$ **do**
 7:          Update the $s_{\text{carry}}$ upon $\{o_{\text{raw}}\}_t$ with Algorithm 1.
 8:          Store trajectory feature $\left\{h_t^{0:L-1}, z_t^{0:L-1}\right\}$ and losses $\mathcal{L}_{\text{dyn}}(\phi), \mathcal{L}_{\text{rep}}(\phi), \mathcal{L}_{\text{rec}}(\phi)$.
 9:      **end for**
10:      {Actor-critic learning }
11:      Stack feature sequence $F \leftarrow \left\{h_{0:T-1}^{0:L-1}, z_{0:T-1}^{0:L-1}\right\}$.
12:      Compute the bootstrapped $\lambda$-return $R_t^\lambda$ and critic loss $\mathcal{L}(\theta)$ with Eq. 9.
13:      View $F$ as a batch of entry points sized $T$.
14:      Open-loop rollout imaginary state-action trajectory of $B$ time-steps starting at entry points batch $F$.
15:      **for** each imaginary trajectory $\{\hat{s}_{0:B-1}, a_{0:B-1}\}$ **do**
16:          Compute the normalized return and actor loss $\mathcal{L}(\psi)$ with Eq. 10.
17:      **end for**
18:      Back propagate losses $\mathcal{L}_{\text{dyn}}(\phi), \mathcal{L}_{\text{rep}}(\phi), \mathcal{L}_{\text{rec}}(\phi), \mathcal{L}(\theta), \mathcal{L}(\psi)$.
19:      Optimize parameters $\phi, \theta, \psi$.
20: **end while**

---

## C. Environment Details

MineDojo agent's initial inventory includes a iron sword, shield, and a full suite of iron armors across all tasks. The maximum number of time-steps for one episode is 1000. For other specifications, see Table 3.

As shown in Table 3, the five Mobs each possess distinct characteristics. Each episode terminates upon timeout or when the agent's health reaches zero, which implies that the agent must not only explore and approach enemies but also learn to evade attacks or defend with a shield. The rich interaction mechanisms thoroughly test the generalization capabilities of RL algorithms.

In MineDojo tasks, the agent is equipped with iron armors and iron sword shield at initialization. We adopt sparse reward from MineDojo at episode termination and dense reward from MineCLIP (Fan et al., 2022). Each MineCLIP reward is

*Table 3.* MineDojo tasks specifications.

| Mobs | Biome | Mob Features | MineCLIP prompt |
|---|---|---|---|
| Spider | extreme hills | Fast movement | combat a spider in night extreme hills with an iron sword, shield, and a full suite of iron armors |
| Shulker | end | Shoots guided bullets which causes floating | combat a shulker in the end with an iron sword, shield, and a full suite of iron armors |
| Wolf | taiga | More agile, group attacks | combat a wolf in taiga with an iron sword, shield, and a full suite of iron armors |
| Skeleton | extreme hills | Accurate ranged attacks with arrows | combat a skeleton in night extreme hills with an iron sword, shield, and a full suite of iron armors |
| Ghast | nether | Flying, ranged attacks with explosive fireball, terrain destruction | combat a ghast in nether with an iron sword, shield, and a full suite of iron armors |

computed of video segment of 16 time-steps, with calculations taking place every 8 frames. In addition, the agent is rewarded for any valid attack and punished for losing health points. The agent is trained for $1 \times 10^6$ environment steps. The image input of the MineCLIP model is $160 \times 256$ pixels, while ResDreamer observes 2x down-sampled images. All experiments can be reproduced with VRAM less than 29 GB.

# D. Additional Results

### D.1. IRIS Baseline

We reproduce IRIS on Minedojo and run an experiment using three random seeds and the official default hyperparameters. The only necessary changes were:

- Adding support for MineDojo's MultiDiscrete action space (instead of the original flat Discrete used on Atari).

- Setting MineDojo observations to 64×64 (instead of our usual 80×128) to match IRIS's hard-coded input resolution. This might weaken the model's capabilities, but is required for compatibility.

We have trained IRIS (Micheli et al., 2022) under default configuration for 7 days. So far, IRIS has failed to achieve meaningful success even on the easiest Combat Spider task throughout 500K environment steps. Our current conclusion regarding the reproduction of IRIS on Minedojo is that the default configuration is unable to complete the Minedojo task. Further hyperparameter tuning is still required.

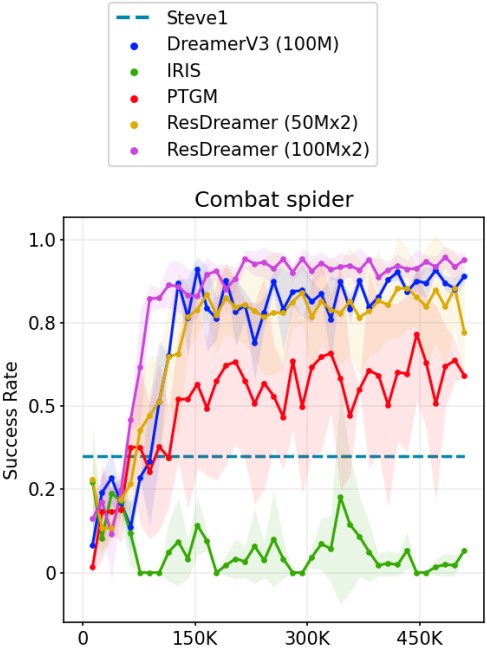

*Figure 7.* Result of IRIS baseline.

## D.2. Main Comparison Bar-chart and Ablation Numerical Result

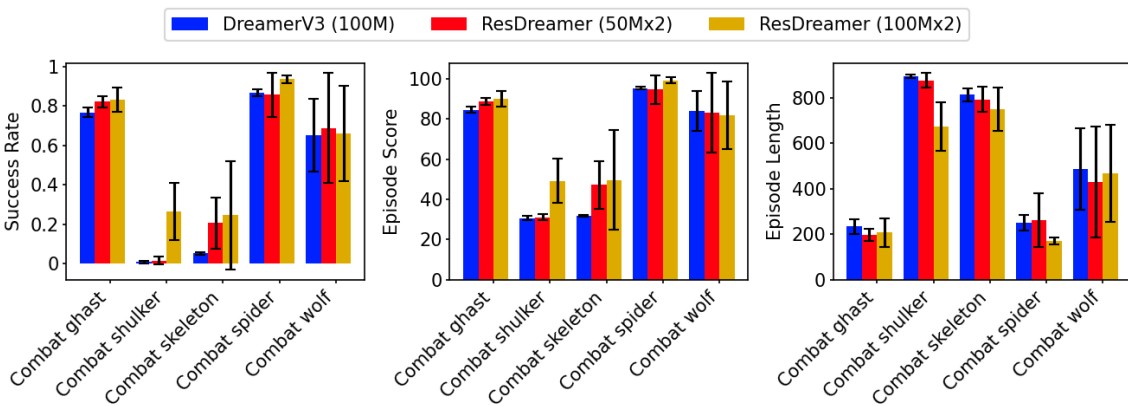

*Figure 8.* Comparisons of success rate (↑), episode score (↑) and episode length (↓) across tasks. It can be seen that ResDreamer achieves higher scores and success rates with fewer steps. Although the ResDreamer (50Mx2) has slightly fewer total parameters than DreamerV3 (100M), it performs better in almost all tasks.

*Table 4.* Ablation result

| Configurations | Hierarchy | Rollout Hint | Residual Connection | Success Rate |
|---|---|---|---|---|
| ResDreamer (50Mx3) | 3 | ✓ | ✓ | 0.776 |
| ResDreamer (50Mx2) | 2 | ✓ | ✓ | 0.727 |
| ResDreamer (Only residual hints) | 2 | | ✓ | 0.563 |
| Dreamer (With roll-out foresight) | 1 | ✓ | | 0.559 |
| ResDreamer (Only rollout hints) | 2 | ✓ | | 0.400 |
| ResDreamer (Heads conditioned on all) | 2 | ✓ | ✓ | 0.377 |

# E. Baseline Introduction

## E.1. Selected Methods

We compare ResDreamer with strong Minecraft RL algorithms, including:

DreamerV3 (Hafner et al., 2025): A model-based RL foundation model. DreamerV3 is trained from scratch without demonstrations and domain knowledge. It generates future latent states recurrently with a non-hierarchical world model.

STEVE-1 (Lifshitz et al., 2023): An finetuned Video Pretraining (VPT) model for open-ended text and visual instructions following. It is post trained through self-supervised behavioral cloning. We test its zero-shot text instructions following performance in MineDojo tasks.

PTGM (Yuan et al., 2024): A hierarchical approach integrating a high-level task goal generation strategy and a low-level goal-conditioned RL strategy. The high-level goal strategy is pretrained on large-scale, task-agnostic datasets, while the low-level strategy is learned online through RL. We utilize the open-source upper-layer strategy parameters of PTGM and evaluate its online training performance on MineDojo tasks using the default configuration of PTGM code-base.

## E.2. Unselected Methods

We provide introductions of other strong Minecraft agents and the reasons we do not compare ResDreamer with them.

LS-Imagine (Li et al., 2024a): An MBRL method that achieves arbitrary time-span reasoning through dual-branch prediction. It is based on DreamerV3, but it supports long-term prediction by simulating jumping to the vicinity of navigation targets through cropping observation. However, combat missions are different from navigation and exploration. Factors such as terrain, enemy reactions, etc. have a significant impact on the expected return, and cutting the images disrupts the data distribution. For instance, it is not reasonable to jump to flying enemies like ghasts by cropping the image.

Voyager (Wang et al., 2023a), JARVIS-1 (Wang et al., 2023b), MC-Planner (Wang et al., 2023c), RL-GPT (Liu et al., 2024): Open-Ended embodied agents that integrates RL with LLM. They adopt heterogeneous hierarchical models, leveraging the prior knowledge of LLMs to achieve task decomposition, long-term planning, code as strategy, and lifelong skill accumulation. Their focus lies in the integration and interaction methods between LLMs and RL, emphasizing the evaluation of an agent's efficiency in accumulating atomic skills and activating technological milestones. Our proposed ResDreamer is a model-based RL foundation model, focusing on evaluating the data efficiency, scalability, and interpretability. ResDreamer can work together with all kinds of upper layer LLMs as a more powerful RL algorithm.

ROCKET-2 (Cai et al., 2025a), ROCKET-3 (Cai et al., 2025b) SkillDiscovery (Deng et al., 2025), JarvisVLA (Li et al., 2025b): Open-world VLA agents powered by imitation learning (IL) and prior knowledge of visual foundation model such as SAM (Kirillov et al., 2023). VLA agents focus on following open instructions within a broader range of atomic skills and their combinations. However, ResDreamer is a MBRL foundation model trained without any prior knowledge. ResDreamer focuses on developing a task-agnostic and domain general hierarchical world model method.

