# OpenReview forum: "Self-supervised Hierarchical Visual Reasoning with World Model"
_ICML.cc/2026/Conference — ICML 2026 regular_

### Official Review · Reviewer_1yRg · 2026-03-09

**Soundness:** 4
**Presentation:** 3
**Significance:** 4
**Originality:** 4
**Overall Recommendation:** 6
**Confidence:** 4

**Summary:**

This paper introduces ResDreamer, a predictive-coding-motivated hierarchical world-model method in which higher layers learn residual dynamics of lower layers and use those residual predictions to modulate visual foresight. The central claim is that useful reasoning representations for open-ended 3D environments do not need photorealistic predictions, but rather compact, task-relevant signals that emphasize surprising or informative future structure. Experiments on MineDojo combat tasks suggest performance that matches or surpasses DreamerV3 on the evaluated settings, with the residual hierarchy presented as the main reason for the gains.

**Compliance With Llm Reviewing Policy:**

Affirmed.

**Final Justification:**

My concerns have been adequately addressed. The authors’ response clarified both of my main points. In particular, the added rollout/foresight visualization makes the residual-modulated predictions much easier to interpret, and the discussion of the sample-efficiency versus compute-time tradeoff gives a clearer practical picture of the method. I also appreciated the thoughtful response regarding temporal hierarchy as a promising orthogonal extension.

Given the rebuttal, I am increasing my score from 5 to 6 (Strong Accept). I believe this paper is technically strong, original, and valuable to the community, and I support its publication.

**Key Questions For Authors:**

1. Would it make sense to introduce a temporal hierarchy across layers, similar to THICK, so that upper layers can learn more abstract event-level concepts rather than only residual refinements at the same timescale?

**Limitations:**

yes

**Strengths And Weaknesses:**

## Strengths
- The motivation is strong and well aligned with predictive-coding ideas; the residual hierarchy gives the method a clear conceptual identity beyond a generic hierarchical world model.
- The reported performance is meaningful because it matches or surpasses DreamerV3, which is a strong and relevant baseline for this setting.
- The ablations are convincing and clearly demonstrate the usefulness of the residual connections.

## Weaknesses
- The paper would benefit from a clearer rollout visualization in the main paper. The appendix visualizations are currently a bit hard to parse, which makes it harder to build intuition for what the residual-modulated foresight is actually capturing and how it differs from more standard rollouts.
- The efficiency discussion should better separate sample efficiency from compute-time efficiency. Table 1 reports substantially higher training time than DreamerV3 (6.2h for DreamerV3 vs 12.3h for ResDreamer 50M×2 and 14.5h for ResDreamer 100M×2), so the practical cost/benefit tradeoff is not fully clear.

---

> ### Author Rebuttal · Authors · 2026-03-29
>
> Dear Reviewer n1aZ,
>
> We appreciate your insightful remarks. It is encouraging to see that your summary of the paper's central claim is highly consistent with our own perspective. We hope that our response will address your concerns.
>
> > W1: Add a clearer rollout visualization to build intuition about what the residual-modulated foresight is actually capturing and how it differs from more standard rollouts.
>
> We appreciate this suggestion. We have added a clear and organized visualization for residual-modulated foresight (available at [https://anonymous.4open.science/r/ResDreamer-F33C/img/ghast_traj.jpg](https://anonymous.4open.science/r/ResDreamer-F33C/img/ghast_traj.jpg)). The foresight visualization was integrated into the data flow diagrams of the corresponding 2-layer world model, along with qualitative analysis. We are confident that it offers a more intuitive global picture and key details of our method.
>
> > W2: Concern about substantially higher training time (6.2h for DreamerV3 100M vs. 12.3h for ResDreamer 50M×2) and not fully clear practical cost/benefit tradeoff.
>
> Admittedly, ResDreamer is more computationally intensive than DreamerV3. We argue that the environmental interaction data acquisition is more often the primary bottleneck. With residual-modulation, we propose a new strategy for compute budget investment—prioritizing hierarchical depth over network width—whose sample- and parameter-efficiency have been empirically verified.
>
> Notably, within our 1M-step budget, the world model is still underfitted (e.g., the visualization at [https://anonymous.4open.science/r/ResDreamer-F33C/img/ghast_traj.jpg](https://anonymous.4open.science/r/ResDreamer-F33C/img/ghast_traj.jpg) shows the model ignoring distant and task-irrelevant objects like a far-off Ghast). A compelling future direction involves incorporating residual-modulation into world model pre-training to assess its cost-benefit tradeoff on diverse downstream metrics once sufficient convergence is achieved.
>
> > Q1: Would it make sense to introduce a temporal hierarchy across layers, similar to THICK, so that upper layers can learn abstract concepts in different time granularity?
>
> Yes, we agree that providing higher-level models with longer-horizon concepts is key to long-term task planning. Techniques like THICK [1], ADM-v2 [2], and Dreamer4 [3] offer orthogonal strengths to embodied world models:
>
> - **THICK [1]**: An additional coarse dynamic pathway is designed to sparsely update its potential state in time, called context. The high-level world model is trained to predict the world state at the next context transition.
> - **ADM-v2 [2]**: In order to reduce bootstrapping prediction to direct prediction, the dynamic model for arbitrary step size prediction is designed to rely only on the initial state and action sequence.
> - **Dreamer4 [3]**: Shortcut Forcing uses short step predictions to bootstrap hidden state predictions for larger step size requests.
>
> While these methods focus on temporal abstraction, our residual-modulation focuses on sensory/representational hierarchy. We believe that integrating these two mechanisms is an important research direction.
>
> [1] Gumbsch, Christian, et al. "Learning hierarchical world models with adaptive temporal abstractions from discrete latent dynamics." *The Twelfth International Conference on Learning Representations*. 2023.
>
> [2] Lin, Haoxin, et al. "ADM-v2: Pursuing Full-Horizon Roll-out in Dynamics Models for Offline Policy Learning and Evaluation." *The Fourteenth International Conference on Learning Representations*. 2026.
>
> [3] Hafner, Danijar, et al. "Training agents inside of scalable world models." *arXiv preprint* [https://arxiv.org/abs/2509.24527](https://arxiv.org/abs/2509.24527)

---

> > ### Author Rebuttal · Reviewer_1yRg · 2026-04-01
> >
> > My concerns have been adequately addressed. The authors’ response clarified both of my main points. In particular, the added rollout/foresight visualization makes the residual-modulated predictions much easier to interpret, and the discussion of the sample-efficiency versus compute-time tradeoff gives a clearer practical picture of the method. I also appreciated the thoughtful response regarding temporal hierarchy as a promising orthogonal extension.
> >
> > Given the rebuttal, I am increasing my score from 5 to 6 (Strong Accept). I believe this paper is technically strong, original, and valuable to the community, and I support its publication.

---

### Official Review · Reviewer_FZS7 · 2026-03-12

**Soundness:** 3
**Presentation:** 2
**Significance:** 3
**Originality:** 3
**Overall Recommendation:** 4
**Confidence:** 3

**Summary:**

The paper seeks to expand upon the Dreamer framework by making the RSSM world model “deeper” – that is, instead of having just one latent representing a frame in a video, have multiple layers of latents. The multi-layer latent state is modeled by a per-layer RSSM.

Furthermore, each layer is tasked with modeling the residual observations i.e., the 0-th layer would simply try to model the raw observations, however, the 1-th layer would try to model the residual/difference between raw observation and the reconstruction from the 0-th layer. Analogously, the 2-th layer would model the remaining residual left after the 1-th layer. This scheme goes on to further upper layers. The posterior in the world model training also relies on what they call the ‘imaginary hint observation’ where, for each layer, the prior is rolled out for a certain number of horizon steps and the resulting roll-out images (residual images) are stacked and fed to the posterior network. While I am cautiously confident that I grasp the main ideas as I described above, the precise details elude me somewhat since the writing in the main paper was not precise enough for me. (See Questions)

The training of the world model and the actor are similar to the original Dreamer.

In experiments, the proposed approach leads to improved performance over Dreamer3 on a range of ‘combat’ tasks. Various ablations are done to support the design choices e.g., the number of hierarchical layers in the world model seem to improve the performance.

**Compliance With Llm Reviewing Policy:**

Affirmed.

**Final Justification:**

I thank the authors for the informative rebuttal, it helps clarify a lot of things! I’d still have preferred a clearer method description. I am fine accepting it but will maintain my score.

**Key Questions For Authors:**

Q. L148 (Right): “updates them with exponential moving average” What does this mean?

Q. How do we get the $\hat{o}$’s in Equation 2? Are they decodings of the latent obtained from posterior or the prior? Same question for Equation 3. The latter specifically makes it difficult to understand what exactly is “imaginary hint observation”. It says “reconstructed open-loop imagination” but that term alone is not a mathematically clear definition of what it is. What is the technical motivation for adding this to the enhanced observation?

Q. In Equation 3, what does it mean (qualitatively speaking) to add residuals from the k-th and (k+1)-th layers? Why not add more layers’ residuals?

Q. I wonder if it’d help by putting a diagram of the graphical model of the world model’s SSM to help the reader easily understand it. The current diagrams are quite confusing. For instance, see Figures 2 and 3 in the RSSM paper [1]

[1] Hafner, Danijar, et al. "Learning latent dynamics for planning from pixels." International conference on machine learning. PMLR, 2019.

**Limitations:**

yes

**Strengths And Weaknesses:**

- [Strength] Sound and well-motivated work. The high-level idea of the approach conforms to the motivation, is novel, and interesting.

- [Strength] Good experimental outcomes, benefits of the proposed approach are visible.
- [Weakness] The writing of the method section and diagrams could have been made clearer. Similarly, the description of the ablation experiments could be made more precise.

- [Strength] Ablations are meaningful and support the design choices.

---

> ### Author Rebuttal · Authors · 2026-03-28
>
> Dear Reviewer FZS7,
>
> We thank you for your thorough and insightful review. We think your questions will help further clarify our approach and refine our presentation.
>
> > Q1: L148 (Right): "updates them with exponential moving average" What does this mean?
>
> The primary purpose of normalizing the residual image is to mitigate distribution drift during training, making it easier for the upper-layer model to extract meaningful features. Specifically, we use an Exponential Moving Average (EMA) to track the running mean and standard deviation of each layer's residual image across pixels. The normalizer maps the residual image to a $0.5 \pm 0.25$ range, so that approximately 95%  (under the assumption of Gaussian distribution) of the residual pixel value samples are mapped to the range [0,1].
>
> > Q2: How do we get the observation reconstruction $ \hat{o}$ in Equation 2? Are they decodings of the latent obtained from posterior or the prior? What exactly is "imaginary hint observation" in Equation 3? What is the technical motivation for adding "imaginary hint observation" to the observation?
>
> In our framework, the source of reconstruction $\hat{o}$ differs between training and imagination:
>
> - **WM training (representation learning)**: Decoding relies on the posterior provided directly by the Encoder. In this process, the posterior is trained to provide the information required to decode new observations together with the recurrent state, while the Predictor is dedicated to inferring the posterior solely using the recurrent state.
> - **WM imagination (latent rollout)**: Decoding relies on the prior generated by the Predictor. Since no environment observations are available, the "imaginary hint observation" in Eq. 3 is derived from the recurrent prediction of prior.
>
> **Motivation:** Technically, this design allows the higher-level WM to focus on **residual sensory information, the  'surprise'** that the lower-level WM failed to capture.
>
> While traditional video generation metrics focus on reducing overall **reconstruction error**, we claim that **modeling open-ended 3D worlds does not necessarily require photorealistic reconstruction, but rather informative task-relevant signals that emphasize unexpected 'surprise'.**
>
> > Q3: In Equation 3, what does it mean (qualitatively speaking) to add residuals from the k-th and (k+1)-th layers? Why not add more layers' residuals?
>
> We chose to add residuals from adjacent layers ($k$ and $k+1$) primarily for:
>
> - **Architectural Simplicity**: This is a design choice that draws on the structure of the classic residual block and the structure of the layers is symmetrical;
> - **Scalability:** The bandwidth consumption for inter-layer communication in the WM system increases linearly with the number of layers.
>
> > Q4: Add a clearer diagram of the world model's SSM to help the reader easily understand it.
>
> We appreciate this suggestion. We have added a comprehensive diagram illustrating the **internal state transitions and data flow** of our 2-layer RSSM at [https://anonymous.4open.science/r/ResDreamer-F33C/img/ghast_traj.jpg](https://anonymous.4open.science/r/ResDreamer-F33C/img/ghast_traj.jpg) . Similar to the graph in the RSSM paper [1], it contains the complete set of hidden states, observations and reasoning chains, where the enhanced observations are exemplified by an actual trajectory.
>
> Thanks again for your precise questions and constructive suggestions. We look forward to further discussions with you.
>
> [1] Hafner, Danijar, et al. "Learning latent dynamics for planning from pixels." International conference on machine learning. PMLR, 2019.

---

> > ### Author Rebuttal · Reviewer_FZS7 · 2026-04-04
> >
> > I thank the authors for the informative rebuttal, it helps clarify a lot of things! I’d still have preferred a clearer method description. I am fine accepting it but will maintain my score.

---

### Official Review · Reviewer_Niy8 · 2026-03-13

**Soundness:** 3
**Presentation:** 3
**Significance:** 3
**Originality:** 3
**Overall Recommendation:** 5
**Confidence:** 2

**Summary:**

This paper proposes ResDreamer, a model based RL method with self-supervised hierarchical visual reasoning representation. In long-horizon settings, the critical factor is producing task-relevant predictive signals that help decision-making under partial observability and dynamics. To explore this, this paper proposes a residual-hierarchy design where each higher layer learns to reconstruct the residual error left by the layer below. The world model is built on PPBs. The residual observation for upper layers is computed from the reconstruction error of the lower layer, normalized with a running mean / variance. Each layer can run open-loop imagination, upper layer residual predictions are added back to lower-layer reconstructions to form an imaginary hint observation. Empirically, the proposed approach achieves improved sample efficiency and parameter efficiency on five MineDojo combat tasks.

**Compliance With Llm Reviewing Policy:**

Affirmed.

**Final Justification:**

The rebuttal addresses most of my concerns and thus I recommend for accept.

**Key Questions For Authors:**

N/A

**Strengths And Weaknesses:**

Strengths:
1. The proposed core idea is interesting, which predicts residuals of the layer below instead of predicting raw pixels.
2. Strong empirical results on the MineDojo dataset, suggesting the effectiveness of the proposed approach.
3. Detailed ablations support the claim that deeper hierarchies can help. The 3-layer variant (50M×3) improves mean success over 2-layer, supporting the “residual hierarchy scales” claim.
4. The visualization of the constructed observations provides good interpretability and illustrates what the residual-enhanced and imaginary-hint observations look like, which is valuable for debugging and conceptual clarity.
5. This paper is well written and easy to follow.

Weaknesses:
1. The proposed approach is only evaluated on the MineDojo dataset. The generalization of the approach is under investigation.
2. The approach is not compared with other strong non-RSSM world models when tuned appropriately for MineDojo task.
3. The paper motivates error accumulation in pixel prediction, but does not present direct metrics of prediction quality. The qualitative figures are helpful but insufficient to validate.
4. Typo in Figure 3. (Eenvironment -> Environment)

---

> ### Author Rebuttal · Authors · 2026-03-27
>
> Dear Reviewer Niy8,
>
> We thank you for your thoughtful and insightful review. We hope that our response will clarify your query and address your concerns.
>
> > W1: Concern about relatively narrow evaluation and generality beyond Minecraft
>
> In our experiments across **Minecraft and DeepMind Vision Control** tasks, we have validated the residual-modulation mechanism yields significant improvements for RL decision-making within a limited budget of 1M interaction steps.
>
> Limited by the length of the rebuttal, **could you please find the detailed answer in response to W1/Q1 of Reviewer n1aZ**? In conclusion, our method is particularly effective in RL environments with complex inter-agent interaction or non-stationary transitions.
>
> > W2: Concern about no comparison with other strong non-RSSM world models properly tuned for MineDojo tasks
>
> RSSM and Transformer-based models are currently the two most performant paradigms for high-dimensional visual environments. In comparison, our RSSM-based framework, although hierarchical, still provides significant efficiency advantages. Due to the computational efficiency of RSSM—further bolstered by optimized RL infrastructure and JAX JIT compilation—our training is more than an order of magnitude faster (~0.5 days vs. 10+ days for Transformer-based WM IRIS). As is shown in appendix C.2, the preliminary results for the IRIS baseline is that its default configuration doesn't work on Minecraft combat, and tuning its hyper-parameters exceeds our current computational budget. We consider integrating Transformers for enhanced sequence modeling as a direction for future work.
>
> > W3: Concern about no direct metrics of video prediction quality
>
> The central claim of our work is that **modeling open-ended 3D worlds does not necessarily require photorealistic reconstruction, but rather task-relevant signals that emphasize surprising or informative future structure**. This position can be more intuitively demonstrated in our new qualitative analysis, please see https://anonymous.4open.science/r/ResDreamer-F33C/img/ghast_traj.jpg .
>
> While traditional metrics focus on reducing global **reconstruction error (perplexity)**, we focus on **informative prediction signals**. Our results show that providing visual reasoning via residual modulation—specifically transfering "surprise" signals—assists downstream decision-making even when the world model is not fully converged.
>
> We believe the true scaling properties of this mechanism will be best verified in large-scale pre-training. We consider it valuable future work to train hierarchical representations until sufficient convergence, and then test it on a wider range of downstream tasks.
>
> Thanks again for your constructive comments and corrections. We have updated the manuscript and look forward to further discussions with you.

---

> > ### Author Rebuttal · Reviewer_Niy8 · 2026-04-03
> >
> > The rebuttal provides more clarification and analysis, and most of my concerns are addressed.

---

### Official Review · Reviewer_n1aZ · 2026-03-13

**Soundness:** 3
**Presentation:** 3
**Significance:** 2
**Originality:** 2
**Overall Recommendation:** 4
**Confidence:** 1

**Summary:**

This article proposes ResDreamer, a self-supervised hierarchical world model based on residual modeling. By allowing high-level learning of low-level prediction errors and modulating visual foresight with residuals, it forms a high information density, task-related visual inference representation. This method significantly improves the sample efficiency and long-term decision-making ability of reinforcement learning in open worlds (such as Minecraft) with a small parameter scale, without relying on language models or domain priors, and achieves leading performance in high-difficulty tasks.

**Compliance With Llm Reviewing Policy:**

Affirmed.

**Final Justification:**

After considering both the paper and the authors’ rebuttal, I maintain the initail recommendation. The rebuttal addressed my questions but did not substantially change my overall confidence in the evaluation.

The paper is technically sound and proposes a reasonable integration of residual modeling into a Dreamer-style hierarchical world model. The empirical results demonstrate clear performance gains within the evaluated settings. However, due to my limited familiarity with this specific sub-area, I remain uncertain about the degree of originality and the broader significance of introducing residuals in this context.

**Key Questions For Authors:**

1. Generality beyond Minecraft: How does ResDreamer perform in non-Minecraft environments, such as continuous-control robotics or non-vision-dominant tasks, and would positive results in these settings change the authors’ claims about general applicability?

2. Qualitative case analysis: Can the authors provide concrete qualitative case studies (e.g., failure/success trajectories) that explicitly show how residual-modulated visual foresight improves decision-making?

**Limitations:**

Yes.

**Strengths And Weaknesses:**

Strengths:
- The paper introduces a novel Dreamer-style world model that incorporates residual learning into a hierarchical architecture, demonstrating clear algorithmic innovation.
- Extensive experimental results validate the superiority of the proposed method in terms of sample efficiency and performance on challenging tasks.

Weaknesses:
- The experimental evaluation is relatively narrow, focusing primarily on combat tasks in Minecraft, and the generalization of the method to other domains such as real-world robotics or non-vision-centric tasks remains to be demonstrated.
- The experiments lack detailed qualitative case studies that explicitly illustrate how the proposed reasoning representations contribute to improved decision-making.

---

> ### Author Rebuttal · Authors · 2026-03-26
>
> Dear Reviewer n1aZ,
>
> We thank you for your thorough and insightful review. We hope that our response will clarify your query and address your concerns.
>
> > W1,Q1: Concern about relatively narrow evaluation and generality beyond Minecraft
>
> We completely agree that it is essential to further clarify the specific domains where our method yields significant improvements, providing other researchers with a clearer understanding of the contributions and limitations of this work.
>
> - **Source of Performance Gains:** In our experiments across all Reinforcement Learning (RL) tasks, the performance enhancements stem from the predictive information provided by sensory signal reasoning and the more expressive latent space representations constructed therefrom.
> - **Factors in RL challenges**: RL performance is also strongly determined by exploration-exploitation dynamics—specifically, whether the policy improvement can be guided by advantage signals with sufficient signal-to-noise ratio during training. While our method significantly enhances visual reasoning, it is not designed to bypass fundamental RL challenges like extreme reward sparsity or complex credit assignment.
> - **Preliminary conclusions on other RL tasks:** Here are some other RL tasks that we have tried to verify the residual-modulation mechanism but were not mentioned in the manuscript:
>
> |Tasks that residual-modulation yields significant improvements  |Tasks description |Reasons why residual-modulation yields significant improvements  |
> |-|-|-|
> |[Atari Boxing](https://ale.farama.org/environments/boxing/)|hitting the opponent in a boxing ring|proactive counter-agent behavior is task relevant |
> |DeepMind Lab [lasertag_three_opponents_small](https://github.com/google-deepmind/lab/blob/master/game_scripts/levels/contributed/dmlab30/README.md#three-opponents-small)  |play laser tag shooter with 3 rule-driven expert opponents  |proactive counter-agent behavior is task relevant and results in non-stationary environmental dynamics  |
>
>
> |Tasks that residual-modulation is NOT significantly helpful |Tasks description |Reasons why residual-modulation is NOT significantly helpful  |
> |-|-|-|
> |[Atari Freeway](https://ale.farama.org/environments/freeway/)|guide the chicken across lane after lane of busy traffic|exploration under extremely sparse reward feedback is the main bottleneck for RL  |
> |DeepMind Lab [skymaze_irreversible_path_varied](https://github.com/google-deepmind/lab/blob/master/game_scripts/levels/contributed/dmlab30/README.md#irreversible-path-varied](https://github.com/google-deepmind/lab/blob/master/game_scripts/levels/contributed/dmlab30/README.md#irreversible-path-varied)|reach a goal connected by a sequence of platforms placed at different heights without backtracking to a higher platform|long-term planning and credit assignment is the main bottlenecks for RL  |
>
>
> Apart from Minecraft combat and DeepMind Vision Control which benefit from residual-modulation mechanism within a limited budget of 1M interaction, we believe a promising future research direction is to integrate the residual-modulation mechanism into the **pre-training of world models**, as real-world visual signals are significantly richer and contain dynamic objects with vastly different complexities (e.g., object permanence, physics and proactive behaviors).
>
> In conclusion, residual-modulation offers substantial benefits for world model reasoning and representation learning in environments with complex physics dynamics or other proactive agents; however, it is not intended to solve the fundamental challenges of RL exploration-exploitation trade-offs or sparse rewards.
>
> > W2,Q2: provide concrete qualitative case studies that explicitly show how residual-modulated visual foresight improves decision-making
>
> We appreciate this suggestion. We have added a qualitative case study available at [https://anonymous.4open.science/r/ResDreamer-F33C/img/ghast_traj.jpg](https://anonymous.4open.science/r/ResDreamer-F33C/img/ghast_traj.jpg) to explicitly demonstrate how residual-modulated visual foresight improves decision-making. This analysis offers a more intuitive global picture and key details of the method.
>
> Thank you for your insightful questions and constructive suggestions. We hope our response provides a clearer demonstration of our work and satisfactorily addresses your concerns. We look forward to your feedback and are happy to engage in further discussion.

---

> > ### Author Rebuttal · Reviewer_n1aZ · 2026-04-04
> >
> > Thank you to the authors for their response.

---

### Decision · Program_Chairs · 2026-04-30

**Decision:**

Accept (regular)

**Comment:**

The reviewers reached a consensus in favor of acceptance, citing several key strengths:
- Originality: Integrating residual modeling into a Dreamer-style hierarchical architecture is novel, interesting, and well-motivated.
- Performance: The method achieves state-of-the-art sample and parameter efficiency on challenging Minecraft combat tasks.
- Ablation: Extensive ablation studies successfully isolated the contributions of the hierarchical depth and residual connections.

Several things during the discussion phase to highlight:
- Generality: In response to concerns about the narrow evaluation scope (Minecraft), the authors provided preliminary results on other tasks like Atari Boxing and DeepMind Lab, showing that residual modulation is particularly effective when proactive counter-agent behavior is relevant.
- Qualitative Analysis: The authors provided additional qualitative case studies and data flow diagrams, making the foresight predictions much easier to interpret.
- Computational Efficiency: While Reviewer 1yRg noted the higher training time compared to standard DreamerV3, they ultimately accepted the authors' argument that the gain in sample efficiency justifies the increased compute investment in many practical scenarios where data acquisition is the primary bottleneck.

The paper is technically sound, well-written, and offers a non-redundant contribution that is likely to be useful to the ICML community interested in world models. While some reviewers suggested that the true scaling properties would be best verified in larger-scale pre-training, they agreed the current results are solid and meaningful. I recommend the paper for acceptance.